# CONTEXT ATTRIBUTION WITH MULTI-ARMED BANDIT OPTIMIZATION

## ABSTRACT

Understanding which parts of the retrieved context contribute to a large language model's generated answer is essential for building interpretable and trustworthy generative QA systems. We propose a novel framework that formulates context attribution as a combinatorial multi-armed bandit (CMAB) problem. Each context segment is treated as a bandit arm, and we employ Combinatorial Thompson Sampling (CTS) to efficiently explore the exponentially large space of context subsets under a limited query budget. Our method defines a reward function based on normalized token likelihoods, capturing how well a subset of segments supports the original model response. Unlike traditional perturbation-based attribution methods (e.g., SHAP), which sample subsets uniformly and incur high computational costs, our approach adaptively balances exploration and exploitation by leveraging posterior estimates of segment relevance. This leads to substantially improved query efficiency while maintaining high attribution fidelity. Extensive experiments on diverse datasets and LLMs demonstrate that our method achieves competitive attribution quality with fewer model queries.

## 1 INTRODUCTION

Retrieval-Augmented Generation (RAG) has emerged as the de facto method for knowledge-intensive question answering tasks, augmenting Large Language Models (LLMs) with external context to improve factual accuracy and credibility Gao *et al.* (2023b). Despite its effectiveness, ensuring that generated answers are genuinely grounded in the provided context remains challenging. LLMs frequently produce hallucinations or incorporate ungrounded information, making it essential to verify and attribute precisely which context segments are responsible for their responses Gao *et al.* (2023a).

Existing approaches to enhancing attribution primarily follow two paradigms. The first involves training models to explicitly cite context segments during generation Nakano *et al.* (2021); Menick *et al.* (2022); Zhang *et al.* (2024); Huang *et al.* (2024). While such techniques improve self-attribution, their reliability is contingent on the model's internal mechanisms. The second paradigm focuses on post-hoc methods, such as ContextCite Cohen-Wang *et al.* (2024), which systematically perturb or mask context segments and evaluate their impact on the output. Although these methods offer greater fidelity by probing the actual input-output behavior of models, they often incur prohibitive computational costs as they incorporate subroutines like LIME Ribeiro *et al.* (2016) and SHAP Lundberg and Lee (2017), which rely on extensive sampling and surrogate modeling, can become impractical for real-world, long-context scenarios due to their high query overhead.

Motivated by the need for a more efficient yet faithful attribution method, we propose framing context attribution as a Combinatorial Multi-Armed Bandit (CMAB) problem. This perspective treats each context segment as a "bandit arm" whose inclusion or exclusion in the context constitutes a combinatorial action, aiming to identify the subset that best supports the generated answer within a limited query budget. Specifically, we define a reward function based on token-level likelihoods, measuring how closely a given subset of context preserves the model's original output distribution. The problem thus becomes selecting the context subset that maximizes this reward with minimal exploration.

To efficiently navigate this combinatorial space, we introduce Combinatorial Thompson Sampling (CTS), a Bayesian bandit method renowned for balancing exploration and exploitation Wang and Chen

(2018). CTS maintains and updates posterior beliefs about each segment's utility, adaptively guiding queries toward subsets most likely to yield informative outcomes. Unlike exhaustive or uniformly random perturbation strategies, CTS significantly reduces the number of model evaluations required, making it especially suited for long-context applications and scenarios demanding interpretability alongside computational efficiency.

Our key contributions are summarized as follows:

- **New Problem Formulation:** We introduce a novel formulation of segment-level context attribution as a combinatorial multi-armed bandit (CMAB) problem.

- **Combinatorial Thompson Sampling for Attribution:** We propose a Combinatorial Thompson Sampling (CTS) algorithm specifically designed for efficient context attribution. By maintaining posterior estimates over segment importance, CTS adaptively selects informative context subsets under tight query budgets.

- **Extensive Empirical Validation:** We conduct comprehensive experiments on three diverse datasets—SST2 Socher *et al.* (2013), HotpotQA Yang *et al.* (2018) and CNN/Dailimail See *et al.* (2017)—using three open-source LLMs, LLaMA3-8B Grattafiori *et al.* (2024) Qwen3-8B Yang *et al.* (2025) and SmolLM-1.7B Allal *et al.* (2024). Results show that CAMAB consistently delivers competitive or superior attribution performance while requiring fewer model queries than existing baselines.

In summary, our approach offers a scalable, principled, and computationally efficient method for context attribution in generative QA systems. Through extensive empirical evaluations, we demonstrate that our CMAB-based framework achieves attribution fidelity comparable to or exceeding state-of-the-art methods, all while drastically reducing the computational demands inherent in traditional perturbation-based techniques. We believe our work represents a significant step towards more interpretable and trustworthy AI assistants, particularly crucial as applications scale to increasingly complex contexts and high-stakes environments.

## 2 RELATED WORK

### 2.1 PERTURBATION-BASED ATTRIBUTION METHODS

A large number of works on model interpretability uses input perturbations to infer feature importance. Techniques like **LIME** Ribeiro *et al.* (2016) and **SHAP** Lundberg and Lee (2017) interpret model predictions by evaluating the model on perturbations of an input and observing how the output changes. LIME fits a local surrogate model (e.g. a linear model) around the neighborhood of the input to estimate each feature's influence. SHAP uses a game-theoretic approach to approximate Shapley values, which quantify each feature's contribution to the prediction in a way that satisfies fairness axioms. These methods are *model-agnostic* and fairly *faithful* in a local sense, but they are notoriously expensive: they require sampling a large number of perturbations for each instance to obtain stable estimates. This cost grows with input dimensionality, making them difficult to apply to settings like long text sequences without sparing accuracy.

### 2.2 ATTRIBUTION METHODS IN LLM SETTINGS

In the context of large language models (LLMs), token-level attribution faces significant challenges due to (1) the combinatorially large perturbation space induced by extensive input contexts, and (2) substantial computational costs associated with individual model queries. Therefore, effective attribution methods must carefully balance explanation fidelity against computational efficiency. Broadly, three strategies have emerged to tackle these challenges:

(i) **Reducing Perturbation Space**: Methods in this category aggregate tokens into higher-level semantic units, such as phrases, sentences, or paragraphs, effectively decreasing the perturbation space. Different granularity levels inherently capture varying degrees of semantic meaning, naturally supporting hierarchical attribution structures. For example, Chen *et al.* (2020) propose a divide-and-conquer strategy that progressively attributes importance from sentence-level down to token-level in text classification. Similarly, MExGen Paes *et al.* (2024) systematically extends perturbation-based

methods like LIME and SHAP to generative LLMs, efficiently identifying influential text spans in a hierarchical manner. ContextCite Cohen-Wang *et al.* (2024) specifically targets segment-level attribution in generative QA scenarios by using SHAP-based perturbation techniques.

(ii) **Pretrained Global Explainers**: Another strategy involves training a global surrogate model as a pretrained explainer, trading upfront training costs for reduced inference latency during explanations. Examples include FEX Pan *et al.* (2025), which employs policy gradient methods to optimize a Bernoulli surrogate explainer, and FastSHAP Jethani *et al.* (2021), which fits a neural network using pseudo-labels derived from SHAP values. Despite their inference efficiency, these methods demand substantial pretraining resources and extensive datasets, and like typical machine learning models, they often encounter generalization challenges when confronted with out-of-distribution samples.

(iii) **Optimizing Perturbation Strategies**: A relatively nascent direction focuses on explicitly optimizing perturbation strategies to reduce the number of required model queries. Sudhakar *et al.* (2021) leverage heuristics derived from input-to-output gradients to selectively perturb features, whereas Pan *et al.* (2021) propose subsequent perturbations aligned with adversarial attack directions. However, such methods typically require internal model knowledge, including gradients or manifold structures, limiting their applicability to models treated as black boxes. Motivated by this gap, our work introduces a novel perturbation sampling method inspired by multi-armed bandit algorithms, enabling dynamic adjustment of perturbations based solely on observed responses, without necessitating internal model information.

### 2.3 BANDIT AND REINFORCEMENT LEARNING APPROACHES

Feature attribution can be viewed as a task of selecting the most informative subsets of features. The multi-armed bandit and reinforcement learning can be utilized to progressively optimize and sequentially search the subsets with a limited budget of actions. Feature selection via multi-armed bandits has been explored in prior research as a way to dynamically identify important features without evaluating all subsets. For example, Nagaraju (2025) propose a feature selection method that uses an Upper Confidence Bound (UCB) bandit algorithm. This allows the algorithm to rapidly converge to a near-optimal feature set, yielding good predictors with fewer feature evaluations. In NLP tasks, BanditSum Dong *et al.* (2018) treated extractive summarization as a contextual bandit problem: given a document (context), their model learned via policy gradient to pick a sequence of sentences (the "action") that maximizes the summary quality reward (ROUGE score) This is an example of using reinforcement learning to select informative subsets of text. Although BanditSum was focused on training a summarization model, the idea of using reward feedback to guide text segment selection is closely related to our approach for attribution.

## 3 PROPOSED METHOD

In this section, we present CAMAB (Context Attribution with Multi-Armed Bandit), our proposed framework for segment-level context attribution in generative QA settings. We begin by formally defining the *context attribution problem* and introducing the notation used for the QA task. We then formulate the problem as a combinatorial multi-armed bandit (CMAB) optimization and describe our solution based on Combinatorial Thompson Sampling (CTS). Finally, we discuss the probabilistic assumptions behind CAMAB and explain how it addresses the *exploration* vs. *exploitation* trade-off inherent to the combinatorial action space of context segment subsets.

### 3.1 PROBLEM FORMULATION

**Context-Supported Generative QA**   We consider a scenario where an LLM is tasked with answering a question $Q$ using a provided context $C$. The context $C$ consists of $N$ discrete segments (e.g. passages, sentences, or paragraphs), $C = \{s_1, s_2, \ldots, s_N\}$. These could be retrieved documents or knowledge snippets relevant to $Q$. The LLM (denoted by $M$) produces a response $R$ which is a sequence of tokens $R = (r_1, r_2, \ldots, r_T)$, generated autoregressively based on the input $(Q, C)$ with proper prompts. We assume $R$ is the response by $M$ when given the full context $C$. Ideally, every factual claim or detail in $R$ should be grounded in some segment of $C$ (if $C$ indeed contained sufficient information to answer $Q$). Our goal is to attribute the content of $R$ back to segments in $C$.

Formally, we define an attribution vector $\boldsymbol{a}$ that assigns each segment $s_j$ a score $a_j$ reflecting how important $s_j$ was for generating $R$. A higher $a_j$ means segment $s_j$ is more responsible for (or contributed more to) the answer.

**Attribution via Context Subsets**    The central insight we leverage is that we can gauge a segment's importance by observing the model's output quality when that segment is absent. If removing a segment (or group) degrades the model's answer, it indicates those segments are crucial; if the output remains unchanged, they likely were not used. This intuition is formalized via a reward function over context subsets.

Let $S \subseteq C$ denote any subset of the full context segments. We define the supportiveness reward $V(S)$ to measure how well the model's answer $R = (r_1, \ldots, r_T)$ is supported when the model has access only to segments in $S$.

Let $L_t(S) = P_M(r_t \mid Q, S, r_1, \ldots, r_{t-1})$ be the log-likelihood of token $r_t$ under subset $S$, and define the reward:

$$V(S) = \exp \left( \frac{1}{T} \sum_{t=1}^{T} (L_t(S) - L_t(C)) \right) \tag{1}$$

where $L_t(C)$ denote the log-likelihood of token $r_t$ under full contexts.

The above result is then clipped to $[0, 1]$ for stability:

$$V(S) = \min \left( \max \left( V(S), 0 \right), 1 \right). \tag{2}$$

This definition ensures that $V(S) = 1$ when $S$ matches the full context in terms of supportiveness, and $V(S) \approx 0$ when $S$ offers no support.

## 3.2 BANDIT FORMULATION WITH THOMPSON SAMPLING

We cast the context attribution problem as a Combinatorial Multi-Armed Bandit (CMAB), where each context segment $j \in \{1, \ldots, N\}$ is treated as a bandit arm. Unlike classical bandits that pull one arm at a time, our setting allows selecting multiple arms, i.e., context segments, in a single round. Each action corresponds to a subset $S \subseteq \{1, \ldots, N\}$, and yields a reward $V(S)$ reflecting how well the selected context supports the original model response.

We assume a stochastic bandit setting, where each segment $j$ is associated with an unknown latent importance parameter $\theta_j \in \mathbb{R}$, indicating its relevance to the generation task. The goal is to efficiently estimate the $\theta_j$ values and use them to select high-quality subsets under a limited query budget.

To balance exploration and exploitation, we adopt **Combinatorial Thompson Sampling (CTS)** Wang and Chen (2018). CTS maintains a posterior distribution over the importance parameters $\boldsymbol{\theta} = \{\theta_j\}_{j=1}^{N}$ and, at each round $t$, samples a plausible realization $\tilde{\boldsymbol{\theta}}^{(t)}$ from the current posterior. Based on the sampled values, it selects the top-$k$ segments with the highest $\tilde{\theta}_j$ to form the action subset $S_t$. This sampling introduces randomness that promotes exploration, while still guiding selection toward promising segments.

We model each $\theta_j$ as a Gaussian random variable, $\theta_j \sim \mathcal{N}(\mu_j, \sigma_j^2)$, where the mean $\mu_j$ reflects the estimated importance of segment $j$ and the variance $\sigma_j^2$ quantifies uncertainty. We initialize with an uninformative prior:

$$\theta_j \sim \mathcal{N} \left( \frac{1}{|C|}, 1 \right),$$

assuming equal expected contribution from all segments at the start. After each query, the posterior is updated using standard Bayesian inference under a Gaussian–Gaussian conjugate prior, based on the observed reward $V(S_t)$ and the segments included in $S_t$.

This formulation allows us to efficiently navigate the combinatorial action space without exhaustively evaluating all $2^N$ subsets. The CTS algorithm is summarized below.

1. **Sample Reward Estimates:** For each segment $j \in \{1, \ldots, N\}$, draw

$$\tilde{\theta}_j^{(t)} \sim \mathcal{N} \left( \mu_j^{(t-1)}, \sigma_j^{2(t-1)} \right), \tag{3}$$

where $\mu_j^{(t-1)}$ and $\sigma_j^{2(t-1)}$ are the posterior mean and variance at round $t-1$.

2. **Action Selection:** Form subset $S_t$ by taking the top-$p$ proportion of segments with the largest $\tilde{\theta}_j^{(t)}$.

3. **Observe Reward:** Query $M$ with $C_{S_t}$ and compute $v_t = V(S_t)$, where $V(\cdot)$ is the reward in Eq. 1.

4. **Posterior Update:** Assume

$$v_t = \sum_{j \in S_t} \theta_j + \epsilon, \quad \epsilon \sim \mathcal{N}(0, \sigma^2). \tag{4}$$

For $j \in S_t$, update

$$\sigma_j^{2(t)} = \left( \frac{1}{\sigma_j^{2(t-1)}} + \frac{1}{\sigma^2} \right)^{-1}, \quad \mu_j^{(t)} = \sigma_j^{2(t)} \left( \frac{\mu_j^{(t-1)}}{\sigma_j^{2(t-1)}} + \frac{v_t}{\sigma^2} \right). \tag{5}$$

and keep $\mu_j^{(t)} = \mu_j^{(t-1)}$, $\sigma_j^{2(t)} = \sigma_j^{2(t-1)}$ for $j \notin S_t$.

5. **Repeat:** Iterate for $t = 1, \ldots, T_{\max}$, where $T_{\max}$ is the query budget.

The process continues until the query budget $T_{\max}$ is exhausted. At the end of this iterative procedure, we obtain a posterior distribution over each segment's latent importance parameter $\theta_j$. The posterior mean $\mu_j^{(T_{\max})}$ serves as a natural estimate of segment $j$'s attribution score, denoted by $a_j$. We rank segments according to these scores to produce the final attribution ranking.

**Comparison to Traditional Methods** When applying traditional perturbation-based attribution methods such as LIME and SHAP in the context attribution setting, we typically rely on uniformly sampled subsets of context segments to estimate their importance. This uniform sampling is agnostic to previously observed outcomes, which can lead to inefficient use of queries, particularly when some segments are already likely to be irrelevant. In contrast, CAMAB addresses this limitation by maintaining and updating posterior beliefs over segment importance throughout the process. These posteriors guide the sampling of segment subsets in a principled manner via Combinatorial Thompson Sampling. As a result, CAMAB allocates queries toward the most informative segments, enabling more efficient and accurate attribution under tight query budgets.

## 4 EXPERIMENTS

In this section, we demonstrate our CAMAB method via experiments on diverse datasets and large language models, the results are evaluated with two distinct metrics. All experiments are conducted on a computing server with 8 CPU cores and one NVIDIA A100 GPU (80GB).

### 4.1 DATASETS

We evaluate our framework on three representative language generation benchmarks that cover distinct task types and context structures. SST-2 focuses on short, token-level attribution for sentiment classification; HotpotQA targets sentence-level attribution in multi-hop question answering; and CNN/DailyMail emphasizes sentence-level attribution for long-document summarization. For computational feasibility, we randomly sample 500 validation instances from each dataset, balancing coverage with the high cost of attribution methods such as SHAP and ContextCite, which require numerous forward passes and combinatorial ablations.

**SST-2 (Stanford Sentiment Treebank)** Socher *et al.* (2013) SST-2 is a sentence-level sentiment classification benchmark where the task is to determine whether a sentence expresses positive or negative sentiment. We treat individual tokens as the segments of interest.

**HotpotQA** Yang *et al.* (2018) HotpotQA is a multi-hop question answering benchmark requiring reasoning over multiple supporting documents to answer factoid questions. Each instance includes long passages, and the responses are more elaborate than in SST-2. We use sentences as the segments of interest.

**CNN/DailyMail**  See *et al.* (2017) CNN/DailyMail is a large-scale abstractive summarization dataset where the task is to generate concise summaries of news articles. Contexts consist of long documents containing narrative and factual information, and the outputs are multi-sentence summaries. We select sentences as the segments of interest.

## 4.2  MODELS

To evaluate the generality and robustness of our context attribution framework, we conduct experiments with three recent open-source large language models that differ in size, training corpus, and design philosophy. This diversity enables us to test the effectiveness of our method across a range of LLM architectures.

**LLaMA-3-8B.**  We use the 8B version of LLaMA 3 Grattafiori *et al.* (2024), a state-of-the-art decoder-only transformer released by Meta AI. Pretrained on a large and diverse corpus with a next-token prediction objective, LLaMA-3 excels at long-context reasoning and produces fluent, high-quality responses, making it a strong high-capacity baseline.

**Qwen3-8B.**  We include Qwen3-8B Yang *et al.* (2025), a decoder-only transformer developed by Alibaba. Trained on large-scale multilingual and multimodal corpora, Qwen3 demonstrates strong cross-lingual generalization and complements LLaMA-3 by representing a distinct pretraining paradigm.

**SmolLM 1.7B.**  We further evaluate SmolLM-1.7B Allal *et al.* (2024), a compact decoder-only model released by Hugging Face. Its limited capacity makes it more prone to hallucinations and less consistent on long-context or complex reasoning tasks. Including SmolLM allows us to assess attribution robustness under constrained model capacity, a setting relevant for cost-sensitive or edge deployments.

Overall, these three models provide complementary testbeds: LLaMA-3-8B and Qwen3-8B serve as strong high-capacity models, while SmolLM-1.7B highlights the challenges of attribution in low-capacity environments.

## 4.3  SETTINGS AND BASELINES

We compare CAMAB against three representative post-hoc attribution baselines, each reflecting a different strategy for identifying influential context segments in generative language model outputs. All methods are evaluated under constrained query budgets, and we control for total LLM calls across methods to ensure fair comparison.

**CAMAB (Ours)**  We set the subset ratio (i.e., portion of selected segments per query) to top-$p = 0.5$ and run the algorithm for $T = 60$ rounds. The observation noise variance is fixed at $\sigma^2 = 1$.

**SHAP**  SHAP Lundberg and Lee (2017) is a widely adopted model-agnostic explainer grounded in Shapley values from cooperative game theory. In our implementation, we adopt the KernelSHAP variant at the segment level. To reduce computational cost, we use a single fully masked context as the reference baseline and limit the number of perturbed samples to 60. The reward signal is selected as the average log-likelihood of the response tokens.

**ContextCite**  ContextCite Cohen-Wang *et al.* (2024) performs context attribution by measuring the average log-odds change in the original response when subsets of context segments are ablated. To identify relevant segments, it employs LASSO regression Tibshirani (1996), which encourages sparsity and effectively filters out non-informative context. To control query cost, we limit the number of ablated subsets to 60.

**Leave-One-Out**  The Leave-One-Out method attributes importance by removing one segment at a time and measuring the effect on the model's output likelihood. We ablate each of the $N$ segments individually and compute the drop in average log-probability in response tokens. This method

Table 1: Evaluation results on LLaMA-3-8B. Top-k means the specific evaluation is done with top k attributed segments removed.

| Dataset | Metrics | Top-$k$ | CAMAB | ContextCite | SHAP | Leave-One-Out |
|---|---|---|---|---|---|---|
| SST2 | Log-Prob Drop ↑ | $k = 1$ | **0.589** | 0.448 | 0.552 | 0.530 |
| | | $k = 3$ | 0.786 | 0.692 | **0.789** | 0.703 |
| | | $k = 5$ | **0.990** | 0.866 | 0.963 | 0.824 |
| | BERTScore ↓ | $k = 1$ | **0.693** | 0.729 | 0.707 | 0.720 |
| | | $k = 3$ | **0.635** | 0.682 | 0.662 | 0.697 |
| | | $k = 5$ | **0.593** | 0.636 | 0.604 | 0.664 |
| HotpotQA | Log-Prob Drop ↑ | $k = 1$ | 0.544 | **0.584** | 0.494 | 0.570 |
| | | $k = 3$ | 0.666 | **0.736** | 0.640 | 0.639 |
| | | $k = 5$ | 0.692 | **0.774** | 0.683 | 0.670 |
| | BERTScore ↓ | $k = 1$ | **0.575** | 0.609 | 0.627 | 0.615 |
| | | $k = 3$ | **0.499** | 0.575 | 0.588 | 0.542 |
| | | $k = 5$ | **0.502** | 0.529 | 0.575 | 0.540 |
| CNN/DailyMail | Log-Prob Drop ↑ | $k = 1$ | 0.383 | 0.341 | 0.348 | **0.414** |
| | | $k = 3$ | 0.865 | 0.792 | 0.851 | **0.868** |
| | | $k = 5$ | 1.100 | 1.093 | **1.204** | 1.150 |
| | BERTScore ↓ | $k = 1$ | **0.622** | 0.662 | 0.623 | **0.622** |
| | | $k = 3$ | 0.541 | 0.530 | **0.506** | 0.531 |
| | | $k = 5$ | **0.451** | 0.465 | 0.460 | 0.466 |

requires exactly $N$ queries per instance, hence will incur more model queries in long-context cases such as in CNN/DailyMail Dataset.

## 4.4 EVALUATION METRICS

To assess the effectiveness of our context attribution method, we adopt two evaluation metrics: *Top-k Log-Probability Drop* and *BERTScore Consistency*.

**Top-$k$ Log-Probability Drop.**  Cohen-Wang *et al.* (2024) This metric evaluates how much the average log-likelihood of the original response $R = (r_1, \ldots, r_T)$ degrades when the top-$k$ most attributed context segments are removed according to a method $\tau$. Let $S_{\text{top-}k}(\tau)$ denote the context subset that excludes the $k$ segments with the highest attribution scores. The Top-$k$ log-probability drop is defined as:

$$\text{Top-}k\text{-drop} = \frac{1}{T} \sum_{t=1}^{T} \log P_M(r_t \mid Q, C, r_{<t}) - \frac{1}{T} \sum_{t=1}^{T} \log P_M(r_t \mid Q, S_{\text{top-}k}(\tau), r_{<t}) \quad (6)$$

Here, $P_M$ is the token-level likelihood under the language model $M$. A larger drop implies that the removed segments were more supportive of the generation, indicating more accurate attribution.

**BERTScore Consistency.**  This metric evaluates attribution fidelity by measuring the semantic difference between the original response $R = (r_1, \ldots, r_T)$, generated using the full context $C$, and the response $R'$, generated using the perturbated context $S_{\text{top-}k}(\tau)$. We compute the BERTScore (Zhang *et al.*, 2019) between the two responses as:

$$\text{BERTScore} = \text{BERTScore}(R', R) \quad (7)$$

A lower BERTScore indicates a greater semantic shift caused by the ablation, suggesting that the removed segments were more influential. Thus, lower values reflect more accurate attribution.

## 4.5 RESULTS

From Table 1, for the LLaMA-3-8B model, CAMAB achieves either the best or second-best results across nearly all metrics. On HotpotQA and CNN/DailyMail, while CAMAB remains competitive, it does not consistently surpass other baselines in log-probability drop. We attribute this to the

Table 2: Evaluation results on Qwen3-8B. Top-k means the specific evaluation is done with top k attributed segments removed.

| Dataset | Metrics | Top-$k$ | CAMAB | ContextCite | SHAP | Leave-One-Out |
|---|---|---|---|---|---|---|
| SST2 | Log-Prob Drop ↑ | $k = 1$ | 1.000 | 0.328 | 1.010 | **1.359** |
| | | $k = 3$ | 1.868 | 0.905 | **2.107** | 2.002 |
| | | $k = 5$ | 2.621 | 1.441 | **2.713** | 2.166 |
| | BERTScore ↓ | $k = 1$ | 0.792 | 0.868 | 0.816 | **0.707** |
| | | $k = 3$ | 0.666 | 0.744 | 0.689 | **0.617** |
| | | $k = 5$ | **0.596** | 0.679 | 0.626 | 0.618 |
| HotpotQA | Log-Prob Drop ↑ | $k = 1$ | 0.780 | 0.536 | 0.782 | **0.830** |
| | | $k = 3$ | **1.126** | 0.786 | 1.018 | 0.973 |
| | | $k = 5$ | **1.179** | 0.876 | 1.106 | 1.060 |
| | BERTScore ↓ | $k = 1$ | **0.631** | 0.755 | 0.668 | 0.642 |
| | | $k = 3$ | 0.591 | 0.632 | 0.607 | **0.589** |
| | | $k = 5$ | **0.526** | 0.597 | 0.586 | 0.564 |
| CNN/DailyMail | Log-Prob Drop ↑ | $k = 1$ | 0.764 | 0.552 | 0.575 | **0.897** |
| | | $k = 3$ | 1.716 | 1.194 | 1.311 | **1.779** |
| | | $k = 5$ | 2.016 | 1.562 | 1.775 | **2.157** |
| | BERTScore ↓ | $k = 1$ | 0.672 | **0.659** | 0.689 | 0.674 |
| | | $k = 3$ | **0.530** | 0.571 | 0.554 | 0.533 |
| | | $k = 5$ | **0.466** | 0.520 | 0.504 | 0.485 |

Table 3: Evaluation results on SmolLM-1.7B. Top-k means the specific evaluation is done with top k attributed segments removed.

| Dataset | Metrics | Top-$k$ | CAMAB | ContextCite | SHAP | Leave-One-Out |
|---|---|---|---|---|---|---|
| SST2 | Log-Prob Drop ↑ | $k = 1$ | 0.407 | 0.381 | **0.414** | 0.408 |
| | | $k = 3$ | 0.523 | 0.498 | **0.534** | 0.523 |
| | | $k = 5$ | 0.572 | 0.554 | **0.588** | 0.569 |
| | BERTScore ↓ | $k = 1$ | 0.377 | 0.432 | **0.371** | 0.384 |
| | | $k = 3$ | 0.354 | 0.404 | **0.342** | 0.385 |
| | | $k = 5$ | 0.355 | 0.328 | **0.324** | 0.357 |
| HotpotQA | Log-Prob Drop ↑ | $k = 1$ | **0.280** | 0.236 | 0.246 | 0.246 |
| | | $k = 3$ | 0.390 | 0.346 | 0.366 | **0.391** |
| | | $k = 5$ | **0.419** | 0.376 | 0.417 | 0.418 |
| | BERTScore ↓ | $k = 1$ | **0.422** | 0.460 | 0.427 | 0.457 |
| | | $k = 3$ | **0.332** | 0.435 | 0.386 | 0.382 |
| | | $k = 5$ | **0.313** | 0.378 | 0.341 | 0.341 |
| CNN/DailyMail | Log-Prob Drop ↑ | $k = 1$ | 0.549 | 0.304 | 0.543 | **0.562** |
| | | $k = 3$ | 0.868 | 0.571 | **0.914** | 0.903 |
| | | $k = 5$ | 1.091 | 0.706 | **1.101** | 1.061 |
| | BERTScore ↓ | $k = 1$ | 0.215 | 0.339 | 0.283 | **0.169** |
| | | $k = 3$ | **0.057** | 0.210 | 0.133 | 0.133 |
| | | $k = 5$ | **0.026** | 0.112 | 0.088 | 0.103 |

long-context nature of these datasets, where longer and compositional responses make token-level *log-probability drop* less reliable as an attribution signal. In contrast, BERTScore captures semantic shifts more faithfully, and CAMAB consistently yields the lowest values across almost all $k$, underscoring its strength in identifying semantically influential segments.

From Table 2, for the Qwen3-8B model, CAMAB outperforms both SHAP and ContextCite across nearly all datasets and metrics. Its performance is broadly comparable to Leave-One-Out, which benefits from exhaustive single-segment ablations but incurs a substantially higher query cost.

From Table 3, we observe that CAMAB maintains competitive performance on the HotpotQA and CNN/DailyMail datasets when using the SmolLM-1.7B model. However, it does not consistently outperform baselines on the SST2 dataset. Notably, the BERTScores across all attribution methods

are substantially lower compared to those obtained with the LLaMA-3-8B and Qwen3-8B models, suggesting a general drop in semantic consistency.

We hypothesize that this discrepancy stems from the lower capacity and robustness of the SmolLM-1.7B model. Due to its reduced parameter count, the model is more sensitive to input perturbations (even when non-relevant context segments are modified), which can lead to disproportionately large changes in its responses. As a result, the attribution quality degrades across all methods, including CAMAB. These findings suggest that *reliable attribution is more challenging on less robust models* and highlight the importance of model stability for effective post-hoc explanation.

## 4.6 ATTRIBUTION WITH LIMITED QUERY BUDGETS

In an ideal setting with unlimited query budgets, exhaustive attribution methods such as SHAP and ContextCite can explore the full $2^N$ space of context segment subsets. However, in realistic scenarios, query budgets are often constrained due to the high computational cost of LLM inference. Under such conditions, it is essential to identify attribution methods that achieve high-quality results with limited queries. We therefore conduct experiments using LLaMA-3-8B on three datasets under restricted query budgets, comparing CAMAB against ContextCite and SHAP. Leave-One-Out is excluded since it requires a fixed number of queries by design. As shown in Table 4, CAMAB not only outperforms ContextCite and SHAP in most cases, but also shows more pronounced improvements as the query budget increases from $s = 20$ to $s = 60$. These results support our claim that CAMAB is particularly well suited for low-budget attribution settings, delivering competitive or superior attribution quality while reducing the number of required model queries.

Table 4: BERTScore performance (lower is better) of **CAMAB**, **ContextCite**, and **SHAP** under varying query budgets ($s = 20, 40, 60$) across datasets using the LLaMA3-8B model. Each $s$ denotes the total number of LLM calls (queries) used for the attribution method. Bold indicates the best performance for each row.

| Dataset | Top-$k$ | CAMAB | | | ContextCite | | | SHAP | | |
|---|---|---|---|---|---|---|---|---|---|---|
| | | $s = 20$ | $s = 40$ | $s = 60$ | $s = 20$ | $s = 40$ | $s = 60$ | $s = 20$ | $s = 40$ | $s = 60$ |
| SST2 | $k = 1$ | 0.743 | 0.719 | **0.693** | 0.757 | 0.741 | 0.729 | 0.705 | 0.708 | 0.707 |
| | $k = 3$ | 0.725 | 0.663 | **0.635** | 0.691 | 0.689 | 0.682 | 0.670 | 0.647 | 0.662 |
| | $k = 5$ | 0.666 | 0.617 | **0.593** | 0.657 | 0.631 | 0.636 | 0.674 | 0.612 | 0.604 |
| HotpotQA | $k = 1$ | 0.578 | 0.581 | **0.575** | 0.628 | 0.611 | 0.609 | 0.679 | 0.636 | 0.627 |
| | $k = 3$ | 0.518 | 0.504 | **0.499** | 0.563 | 0.541 | 0.575 | 0.640 | 0.584 | 0.588 |
| | $k = 5$ | 0.524 | 0.509 | **0.502** | 0.563 | 0.536 | 0.529 | 0.661 | 0.576 | 0.575 |
| CNN/Dailymail | $k = 1$ | 0.693 | 0.634 | **0.622** | 0.725 | 0.663 | 0.662 | 0.680 | 0.642 | 0.623 |
| | $k = 3$ | 0.537 | 0.534 | 0.541 | 0.559 | 0.527 | 0.530 | 0.559 | 0.520 | **0.506** |
| | $k = 5$ | 0.504 | 0.479 | **0.451** | 0.479 | 0.473 | 0.465 | 0.512 | 0.489 | 0.460 |

## 5 CONCLUSION

We introduced CAMAB, a novel context attribution framework that formulates attribution as a combinatorial multi-armed bandit (CMAB) problem and applies Combinatorial Thompson Sampling to efficiently explore the exponentially large space of context subsets. Unlike traditional perturbation methods that rely on random or exhaustive sampling, CAMAB dynamically adapts its exploration based on evolving segment relevance, enabling high attribution fidelity under tight query budgets. Empirical results across multiple datasets and models show that CAMAB achieves attribution quality comparable to or better than existing baselines, while requiring fewer model queries and benefiting more from additional budget. These findings highlight CAMAB's potential for real-world, low-resource interpretability scenarios, making it a practical and scalable solution for faithful context attribution in generative QA systems.

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
