# OpenReview forum: "Context Attribution with Multi-Armed Bandit Optimization"
_ICLR.cc/2026/Conference — ICLR 2026 Conference Withdrawn Submission_

### Official Review · Reviewer_YBij · 2025-10-31

**Soundness:** 1
**Presentation:** 2
**Contribution:** 2
**Rating:** 2
**Confidence:** 4

**Summary:**

The paper studies the problem of segment-level context attribution. The authors formulate attribution as a multi-armed bandit problem, where each segment is an arm and an action corresponds to selecting a subset of segments. The reward is defined as the change in token log-likelihood of the original answer when conditioning on a subset versus the full context. The objective is to find high-reward subsets under a limited query budget. The proposed method CAMAB uses Combinatorial Thompson Sampling with Gaussian posteriors per segment. At each round, it samples per-segment utilities, selects the top-p subset, queries the LLM to obtain the reward, and updates the posteriors using a linear semi-bandit observation model. The authors compare CAMAB with baselines such as ContextCite, Leave-One-Out, and SHAP. Experiments are conducted on 3 different tasks using 3 different LLMs of varying sizes.

**Strengths:**

* Context attribution is an important problem for RAG applications.


* The experiments include both token-level and sentence-level attribution.

**Weaknesses:**

* The main motivation of this work is to improve efficiency compared to perturbation-based methods such as LIME and SHAP. However, there already exist efficient variants such as TreeSHAP that mitigate the extensive sampling problem. The authors should discuss and include comparisons with these variants. Moreover, none of the datasets used in the paper involves long-context scenarios, which somewhat weakens the motivation.
* I am not fully convinced by the MAB framing. What is exploration and exploitation in the context attribution setting?
* There is more recent work on context attribution that the authors neither discuss nor compare against [1, 2, 3].
* The results in Tables 1–3 do not clearly show the advantages of CAMAB as the improvements are marginal. The authors should report standard deviations and/or conduct statistical significance tests.

[1] Cohen-Wang, Benjamin, Yung-Sung Chuang, and Aleksander Madry. "Learning to Attribute with Attention." arXiv preprint arXiv:2504.13752 (2025).

[2] Liu, Fengyuan, Nikhil Kandpal, and Colin Raffel. "AttriBoT: A Bag of Tricks for Efficiently Approximating Leave-One-Out Context Attribution." ICLR 2025.

[3] Xiao, Yingtai, et al. "TokenShapley: Token Level Context Attribution with Shapley Value." arXiv preprint arXiv:2507.05261 (2025).

**Questions:**

* Could the paper report runtime comparisons between CAMAB and other baselines?
* Since the Leave-One-Out baseline removes one segment at a time and measures the effect on the model's output likelihood, shouldn't it serve as an upper bound performance when $k = 1$? Could the authors clarify why CAMAB performs better than this baseline?

---

> ### Author Response · Authors · 2025-11-27
> **Thanks for the valuable feedbacks!**
>
> We appreciate the reviewer’s valuable feedback. Below we provide clarifications on the raised concerns.
>
> 1. The main motivation of this work is to improve efficiency compared to perturbation-based methods such as LIME and SHAP. However, there already exist efficient variants such as TreeSHAP that mitigate the extensive sampling problem. The authors should discuss and include comparisons with these variants. Moreover, none of the datasets used in the paper involves long-context scenarios, which somewhat weakens the motivation.
>
> **Regarding the concern about existing efficient SHAP variants such as TreeSHAP:** TreeSHAP is an exact algorithm specifically designed for decision-tree ensembles (e.g., XGBoost, LightGBM). Its efficiency comes from exploiting tree-structure properties, such as split paths. These structures do not exist in transformer-based LLMs, hence alternatives like TreeSHAP is not applicable in LLM settings.
>
> **Regarding the motivation and the use of long-context datasets:** HotpotQA and CNN/DailyMail indeed represent long-context scenarios containing multi-sentence passages (HotpotQA) or full news articles with tens of sentences (CNN/DailyMail). Only SST-2 involves short sentences and is included to test token-level attribution.
>
> 2. I am not fully convinced by the MAB framing. What is exploration and exploitation in the context attribution setting?
>
> In traditional perturbation-based attribution, subsets of segments are sampled uniformly, and each query is independent of all previous ones. This leads to substantial redundancy, especially when many segments are irrelevant. The motivation for the MAB formulation is precisely to leverage **exploration and exploitation** to make perturbation sampling adaptive and query-efficient.
>
> - **Exploration**: querying subsets that include segments whose importance is still uncertain. This helps reduce uncertainty and prevents the algorithm from prematurely ignoring potentially important segments.
> - **Exploitation**: focusing queries on segments that are most likely to be influential. Thereby allocating more budget to validating or refining important segments.
>
> Thompson sampling naturally balances these two behaviors by sampling from the posterior over segment utilities at each round. As a result, CAMAB avoids repeatedly perturbing segments that quickly appear uninformative, leading to improved attribution efficiency under a limited query budget.
>
> 3. There is more recent work on context attribution that the authors neither discuss nor compare against [1, 2, 3].
>
> We appreciate the reviewer for pointing out these recent papers, and we clarify how they relate to our setting.
>
> [1] Cohen-Wang, Benjamin, Yung-Sung Chuang, and Aleksander Madry. "Learning to Attribute with Attention." arXiv preprint arXiv:2504.13752 (2025).
>
> This work proposes a training-based method that learns an attention explainer. Since CAMAB focuses on post-hoc, black-box attribution without additional training, this method is orthogonal to ours.
>
> [2] Liu, Fengyuan, Nikhil Kandpal, and Colin Raffel. "AttriBoT: A Bag of Tricks for Efficiently Approximating Leave-One-Out Context Attribution." ICLR 2025.
>
> AttriBoT is an efficient implementation of the Leave-One-Out (LOO) strategy using a bag of heuristics to accelerate LOO evaluation. Since we already include LOO as a baseline, our evaluation covers the underlying attribution principle of AttriBoT. We will clarify this connection in the revision and add a discussion of AttriBoT in the Related Work section.
>
> [3] Xiao, Yingtai, et al. "TokenShapley: Token Level Context Attribution with Shapley Value." arXiv preprint arXiv:2507.05261 (2025).
>
> TokenShapley targets token-level Shapley explanations, while CAMAB focuses on segment-level attribution under limited queries. The only applicable experiment would only be in the SST2 settings. As this work appeared very recently, we were unable to include it initially; however, we will discuss it in the revised version and consider including experiments where applicable.

---

> > ### Author Response · Authors · 2025-11-27
> > **continued**
> >
> > 4. The results in Tables 1–3 do not clearly show the advantages of CAMAB as the improvements are marginal. The authors should report standard deviations and/or conduct statistical significance tests.
> >
> > We acknowledge that the empirical advantage in some settings in Table 1-3 are marginal, however these do not contradict CAMAB’s advantages in **long-context, budget-constrained attribution scenarios**. Here we provide a more unified explanation of the underlying factors to the observed variations.
> >
> > 1. Effect of response length : *Log-Prob Drop* is a metric reliable for short, concise generations, but can be less informative for long, compositional responses where semantic divergence cannot be faithfully captured by perplexity-style metrics. This explains why no single method consistently dominates this metric for datasets with long responses (HotpotQA, CNN/DailyMail).
> > 2. Effect of context length: SST-2 contains very short contexts. In such settings, the perturbation space is small and can be effectively explored even with simple baselines. Combined with its short responses, this explains why SHAP performs well in *Log-Prob Drop* under SST-2: the search space is inherently easy and the metric is well aligned with the short responses.
> > 3. Effect of model capacity: Lower-capacity models such as SmolLM-1.7B are substantially more sensitive to input perturbations. Even removing non-essential segments can induce disproportionately large changes in the output, lowering attribution quality for *all* methods. This is consistent with common understanding that model robustness is crucial for stable post-hoc attribution.
> >
> > We propose CAMAB not intended to outperform all baselines in all regimes, but to provide a principled and scalable attribution mechanism in scenarios where perturbation spaces are large and query budgets are tight (as shown in Table 4).
> >
> > **Regarding the request for standard deviations and statistical significance tests:** We appreciate the reviewer’s suggestion. Running statistical tests for attribution metrics in LLM settings typically requires dozens of repeated forward passes *per instance* and *per method*. Since each attribution method already requires tens to hundreds of model queries, repeating the full pipeline multiple times exceeds our computational and budget constraints. Nevertheless, we agree that reporting variability is valuable. In the revised version, we will include variance estimates computed across instances, which is feasible without re-running the entire attribution procedure multiple times.

---

### Official Review · Reviewer_3iHs · 2025-11-01

**Soundness:** 3
**Presentation:** 3
**Contribution:** 2
**Rating:** 4
**Confidence:** 4

**Summary:**

The paper proposes CAMAB, a new framework for context attribution in retrieval-augmented generation (RAG) systems with large language models (LLMs). By formulating the attribution task as a combinatorial multi-armed bandit (CMAB) problem and leveraging Combinatorial Thompson Sampling (CTS), the method aims to efficiently identify which context segments most support a model’s generated answer. The approach is evaluated on three datasets (SST2, HotpotQA, CNN/DailyMail) and three LLMs (LLaMA3-8B, Qwen3-8B, SmolLM-1.7B), showing competitive or slightly superior attribution quality with fewer model queries compared to established baselines like SHAP and ContextCite.

**Strengths:**

Novel Formulation: The CMAB framing for context attribution is original and addresses the inefficiency of traditional perturbation-based methods.
Principled Algorithm: The use of Combinatorial Thompson Sampling for adaptive exploration/exploitation is well-motivated and theoretically grounded.
Empirical Evaluation: The experiments are thorough, spanning multiple datasets and models, and use both log-probability drop and BERTScore metrics.
Practical Relevance: Query efficiency is a real concern for LLM applications, and the method is designed with this in mind.
Clear Baseline Comparisons: The paper benchmarks against strong baselines and explores performance under varying query budgets.

**Weaknesses:**

Marginal Empirical Gains: The improvements over existing baselines are generally small, and in some cases, CAMAB is only comparable rather than clearly superior—even under limited query budgets. This raises questions about the practical impact and significance of the contribution.

**Questions:**

Given the marginal improvement over existing prior works, what is key value add to attribution with the proposed approach?

---

> ### Author Response · Authors · 2025-11-27
> **Thanks for the valuable feedbacks!**
>
> We appreciate the reviewer’s valuable feedback. Below we provide clarifications on the raised concerns.
>
> Q: Marginal Empirical Gains: The improvements over existing baselines are generally small, and in some cases, CAMAB is only comparable rather than clearly superior—even under limited query budgets. This raises questions about the practical impact and significance of the contribution.
>
> A: The key advantage of CAMAB lies in query-efficient exploration of exponentially large perturbation spaces. This enables CAMAB to scale to long contexts and tight budgets, which is typical for LLM applications.
>
> We acknowledge that the empirical advantage in some settings in Table 1-3 are marginal, however these do not contradict CAMAB’s advantages in **long-context, budget-constrained attribution scenarios**. Here we provide a more unified explanation of the underlying factors to the observed variations.
>
> 1. Effect of response length : *Log-Prob Drop* is a metric reliable for short, concise generations, but can be less informative for long, compositional responses where semantic divergence cannot be faithfully captured by perplexity-style metrics. This explains why no single method consistently dominates this metric for datasets with long responses (HotpotQA, CNN/DailyMail).
> 2. Effect of context length: SST-2 contains very short contexts. In such settings, the perturbation space is small and can be effectively explored even with simple baselines. Combined with its short responses, this explains why SHAP performs well in *Log-Prob Drop* under SST-2: the search space is inherently easy and the metric is well aligned with the short responses.
> 3. Effect of model capacity: Lower-capacity models such as SmolLM-1.7B are substantially more sensitive to input perturbations. Even removing non-essential segments can induce disproportionately large changes in the output, lowering attribution quality for *all* methods. This is consistent with common understanding that model robustness is crucial for stable post-hoc attribution.
>
> We propose CAMAB not intended to outperform all baselines in all regimes, but to provide a principled and scalable attribution mechanism in scenarios where perturbation spaces are large and query budgets are tight (as shown in Table 4).

---

### Official Review · Reviewer_sFGw · 2025-11-01

**Soundness:** 2
**Presentation:** 2
**Contribution:** 2
**Rating:** 2
**Confidence:** 3

**Summary:**

The paper introduces CAMAB, a framework for context attribution that identifies which retrieved segments most influence an LLM's output. It formulates the attribution task as a CMAB problem, where each context segment is treated as an arm, and uses combinatorial Thompson sampling to efficiently explore the exponentially large space of possible context subsets under limited query budgets. A reward function based on normalized token-likelihood differences measures how well a subset of segments supports the model's original answer. By maintaining Bayesian posteriors over each segment’s latent importance and sampling from these distributions to choose subsets, CAMAB adaptively balances exploration and exploitation, focusing model queries on the most informative context portions. This approach achieves attribution fidelity comparable to or better than traditional perturbation-based methods like SHAP or LIME, while requiring far fewer model evaluations.

**Strengths:**

* The authors address an important problem of contextual attribution
* The experimental sections design demonstrates generality across different levels of compositionality and context length.
It also supports the claim that CAMAB is versatile across diverse generative settings.

**Weaknesses:**

* The combinatorial MAB framing is elegant in theory but effectively reduced to an additive approximation that ignores segment interactions. While CTS provides tractable sampling, it does not truly explore the exponentially large action space; it merely assumes away the combinatorial complexity through a linear independence assumption.
* Although presented as a Bayesian bandit innovation, the algorithm effectively performs an online approximation of SHAP-like marginal attribution. The independence and additivity assumptions make it efficient but at the cost of fidelity: segment dependencies, negations, or redundancy are not properly captured.
* Results in Tables 1–3 show that CAMAB’s advantage is not consistent. The authors evaluate only 500 samples per dataset, randomly drawn from validation sets.
* A more principled alternative would be to frame attribution as a contextual MAB problem, where each context segment's expected reward depends on features derived from the question, retrieval context, and model generation. Such a formulation would allow attribution to adapt dynamically to query-specific semantics, capture complementarity and redundancy between segments, and leverage shared structure for improved sample efficiency. Such modelling would also help to propose different reward-schemes that do not mimic SHAP. For e.g., reward based on semantic alignment between a segment and the generated answer, pproximate influence using gradients etc.

**Questions:**

Kindly refer to weaknesses

---

> ### Author Response · Authors · 2025-11-27
> **Thanks for the valuable feedbacks!**
>
> We appreciate the reviewer’s valuable feedback. Below we provide clarifications on the raised concerns
>
> 1. The combinatorial MAB framing is elegant in theory but effectively reduced to an additive approximation that ignores segment interactions …
> 2. … The independence and additivity assumptions make it efficient but at the cost of fidelity: segment dependencies, negations, or redundancy are not properly captured.
>
> We address these two comments together. We would like to clarify that CAMAB **does not** assume independence among segments and does not discard interaction effects.  In a combinatorial bandit setting, each posterior update is performed at the subset level: whenever two segments co-occur in a sampled subset, their posterior parameters are updated **jointly** based on the subset reward. As a result, any synergistic or complementary interaction that manifests only when segments appear together is naturally reflected in their posterior means. Segments that produce high reward *only when jointly present* will co-occur more frequently under Thompson sampling, and their posterior estimates will increase together.
>
> This mechanism is identical in spirit to Shapley marginalization, where higher-order interactions are implicitly absorbed into the main-effect attributions (for example, the row-sum of the Shapley interaction matrix recovers the first-order SHAP vector). CAMAB thus captures interaction effects implicitly through co-occurrence.
>
> 3. Although presented as a Bayesian bandit innovation, the algorithm effectively performs an online approximation of SHAP-like marginal attribution.
>
> Our contribution specifically targets improving efficiency and addressing the relatively new setting of segment-level attribution for long-context LLMs, while maintaining competitive fidelity. Given the extremely high query cost of LLMs, we argue that algorithmic improvements that reduce budget requirements are nontrivial and practically important.
>
> 4. Results in Tables 1–3 show that CAMAB’s advantage is not consistent. The authors evaluate only 500 samples per dataset, randomly drawn from validation sets.
>
> We acknowledge that the empirical advantage is not fully consistent across all settings, and we have already discussed these observations in Section 4.5. Here we provide a more unified explanation of the underlying factors:
>
> 1. Effect of response length :  *Log-Prob Drop* is reliable for short, concise generations, but can be less informative for long, compositional responses where semantic divergence cannot be faithfully captured by perplexity-style metrics. This explains why no single method consistently dominates this metric for datasets with long responses (HotpotQA, CNN/DailyMail).
> 2. Effect of context length: SST-2 contains very short contexts. In such settings, the perturbation space is small and can be effectively explored even with simple baselines. Combined with its short responses, this explains why SHAP performs well in *Log-Prob Drop* under SST-2: the search space is inherently easy and the metric is well aligned with the short responses.
> 3. Effect of model capacity: Lower-capacity models such as SmolLM-1.7B are substantially more sensitive to input perturbations. Even removing non-essential segments can induce disproportionately large changes in the output, lowering attribution quality for *all* methods. This is consistent with common understanding that model robustness is crucial for stable post-hoc attribution.
>
> These factors jointly explain the observed variations in Tables 1–3, without contradicting CAMAB’s advantages in long-context, budget-constrained attribution scenarios.
>
> For limited number of sampled data, due to the extremely high cost of baselines such as SHAP and ContextCite, scaling baselines to full datasets would be expensive.
>
> 5. A more principled alternative would be to frame attribution as a contextual MAB problem …
>
> We thank the reviewer for this insightful suggestion. We agree that framing context attribution as a contextual MAB is a potential direction. We view this as an inspiring extension of our current work. While our focus in this paper is on developing an efficient framework under strict query budgets, a contextual MAB may require pretraining to formulate an extractor for deriving features from the question, retrieval context, and model generation. This indicates that incorporating contextual features can be compatible with the CAMAB framework and represents a valuable avenue for future research.

---

### Official Review · Reviewer_sEzR · 2025-11-03

**Soundness:** 3
**Presentation:** 3
**Contribution:** 2
**Rating:** 4
**Confidence:** 4

**Summary:**

This paper presents a framework for interpreting generative QA models by identifying context segments that influence a large language model’s responses. The proposed method formulates context attribution as a combinatorial multi-armed bandit problem and applies Combinatorial Thompson Sampling to efficiently explore context subsets under limited queries. A reward function based on normalized token likelihoods quantifies each segment’s contribution. Unlike perturbation-based methods adaptively balances exploration and exploitation through posterior relevance estimates, achieving high attribution fidelity with significantly fewer model queries.

**Strengths:**

The authors reformulate segment-level context attribution as a combinatorial multi-armed bandit problem.

**Weaknesses:**

The proposed context attribution method does not fundamentally differ from perturbation- or mask-based approaches; it mainly introduces improvements in sampling or subset partitioning. Moreover, as acknowledged by the authors, traditional methods can achieve better performance when computational resources are sufficient. Hence, the theoretical guarantees and advantages of the proposed method require further clarification.
In the experimental section, the dataset used by the authors appears to be small, which limits the ability to accurately assess the model’s performance and relevance to current benchmarks.
As a reminder, that many formulas are difficult to follow and lack adequate annotations or explanations, making it challenging for readers to fully understand the theoretical derivations.

**Questions:**

Please refer to the Weaknesses.

---

> ### Author Response · Authors · 2025-11-26
> **Thanks for the valuable feedbacks!**
>
> We appreciate the reviewer’s valuable feedback. Below we provide clarifications on the raised concerns
>
> 1. The proposed method does not fundamentally differ from perturbation- or mask-based approaches.
>
> While our method follows the general perturbation–observe–aggregate paradigm, its novelty lies in *how* perturbations are selected and *how* evidence accumulates. Existing methods differ primarily in (i) how attribution is aggregated (e.g., SHAP vs. LIME), (ii) how efficiency is improved (e.g., TreeSHAP vs. KernelSHAP), and (iii) how they adapt to different contexts (e.g., ContextCite vs. SHAP).
>
> Our contribution specifically targets (ii) improving efficiency and (iii) addressing the relatively new setting of segment-level attribution for long-context LLMs, while maintaining competitive fidelity. Given the extremely high query cost of LLMs, we argue that algorithmic improvements that reduce budget requirements are nontrivial and practically important.
>
> 2. Traditional methods can achieve better performance when computational resources are sufficient.
>
> We acknowledge this and explicitly stated it in the paper. With unlimited budget, most perturbation-based approaches converge to similar estimates, making efficiency less critical. However, the LLM attribution setting is inherently budget-constrained and expensive. Addressing attribution under tight query budgets is therefore a core challenge that prevents classical methods from being used in practice. Our method is designed precisely for such practical constraints.
>
> 3. The theoretical guarantees and advantages require further clarification.
>
> Thompson Sampling for combinatorial semi-bandits enjoys a known regret bound of $O(m\sqrt{T\log T})$ where m is the number of arms, T is number to pulling rounds. ( https://arxiv.org/pdf/1803.04623).
>
> The key advantage is the posterior-guided, greedy-style sampling that extracts more information from fewer observations, enabling efficient attribution even when the action space is exponentially large.
>
> 4. The dataset appears small and may limit evaluation reliability.
>
> We selected SST-2, HotpotQA, and CNN/DailyMail because they represent distinct attribution granularities: token-level, short-sentence–level, and long-sentence–level.
>
> For LLM-based attribution, it is standard practice to subsample for evaluation due to the extremely high cost of baselines such as SHAP and ContextCite. We follow this common practice by using 500 randomly selected examples per dataset. Scaling baselines to full datasets would be infeasible.
>
> 5. Many formulas are difficult to follow due to limited explanation.
>
> We appreciate this comment. We are happy to revise the manuscript to improve clarity and will add more annotations, intuition, and step-by-step explanations. If the reviewer can point out specific equations, we will refine those sections accordingly.

---

### Note · Authors · 2026-01-05

I have read and agree with the venue's withdrawal policy on behalf of myself and my co-authors.